# Differences in Odor Identification in Early-Onset and Late-Onset Depression

**DOI:** 10.3390/brainsci12020276

**Published:** 2022-02-16

**Authors:** Meiling Liu, Ben Chen, Xiaomei Zhong, Min Zhang, Qiang Wang, Huarong Zhou, Zhangying Wu, Le Hou, Qi Peng, Si Zhang, Minfeng Yang, Gaohong Lin, Yuping Ning

**Affiliations:** 1The Affiliated Brain Hospital of Guangzhou Medical University, Guangzhou 510000, China; lmeiling630@163.com (M.L.); chenbenpielo@sina.com (B.C.); lovlaugh@163.com (X.Z.); 13670881039@163.com (M.Z.); wqhadron@163.com (Q.W.); dushuren0607@163.com (H.Z.); janewzy@live.cn (Z.W.); holyhou@163.com (L.H.); simppy090@163.com (Q.P.); zhangsi19972020@outlook.com (S.Z.); ymfmefil@163.com (M.Y.); lingaohong125@126.com (G.L.); 2The First School of Clinical Medicine, Southern Medical University, Guangzhou 510000, China; 3Guangdong Engineering Technology Research Center for Translational Medicine of Mental Disorders, Guangzhou 510000, China

**Keywords:** late life depression, odor identification, olfactory dysfunction, cognitive impairment, mediating effect

## Abstract

(1) Background: Odor identification (OI) dysfunction is a potential predictor of developing dementia in late life depression (LLD). However, it is not clear whether patients with early onset depression (EOD) and late onset depression (LOD) may exhibit different OI dysfunctions. The aim of this study was to compare OI between EOD patients and LOD patients and its relationship with cognitive function. (2) Methods: A total of 179 patients with LLD and 189 normal controls were recruited. Participants underwent clinical assessment, olfactory testing, and comprehensive neuropsychological assessment. The OI scores of EOD patients and LOD patients were compared, and correlation analyses and mediation analyses were used to explore the relationship between OI and cognition. (3) Result: LOD patients exhibited lower OI scores than EOD patients and normal controls (NCs). Additionally, the LOD patients exhibited a higher percentage of OI dysfunction than the EOD patients. Moreover, OI scores were associated with global cognition, memory, language, and visuospatial ability in the EOD group (*p* < 0.05) but were not associated with any cognitive score in the LOD patients (*p* > 0.05). Finally, the scores of the Auditory Verbal Learning Test Immediate recall and Boston Naming Test exhibited a partially mediating effect on the difference in OI scores between the EOD and LOD patients. (4) Conclusions: LOD patients exhibited worse OI than EOD patients, and their difference in OI was mediated by their memory and language function.

## 1. Introduction

Late life depression (LLD) is one of the leading risks of disability and dementia in the geriatric population [1], with approximately 3.68–4.60% of older adults being diagnosed with depression each year [2]. LLD can be stratified by age into early-onset depression (EOD) and late-onset depression (LOD) according to whether the first depressive episode occurs before or after age 60 [3,4]. Accumulating studies suggest that the underlying aetiology, clinical presentation, course, and response to treatment are different between EOD and LOD [5,6]. Specifically, EOD is primarily associated with adverse life events and genetic susceptibility, whereas LOD is more associated with microvascular dysfunction, stroke, neurodegeneration, and other pathological ageing processes [7,8]. Additionally, LOD is associated with a higher risk of developing Alzheimer’s disease (AD) compared with EOD [9], and many predictors of AD are also more associated with LOD, such as more hippocampal atrophy [10,11], glucocorticoid dysregulation [12,13], nerve growth factor deficiency [14], elevated Aβ burden [15], more white matter hypersignal [16], and higher inflammatory levels [17,18,19].

Odor identification (OI) dysfunction is known to be a noninvasive and inexpensive predictor for developing AD, with accuracy comparable with that of cerebrospinal fluid or neuroimaging biomarkers [20,21]. Longitudinal studies showed that OI dysfunction predicted a faster cognitive decline and a higher rate of AD conversion in cases of amnestic mild cognitive impairment (aMCI) and old adults with normal cognition [22,23]. Additionally, our previous studies suggested that compared with LLD patients with intact OI, LLD patients with OI dysfunction exhibited more severe cognitive impairment and structural and functional brain abnormalities, which suggested that OI impairment may also serve as a marker for predicting the conversion from LLD to dementia [24,25,26]. However, it is not clear whether patients with EOD and LOD may exhibit different OI, which is extremely important for the rational use of olfaction to predict dementia risk.

There are many studies indicating that the OI in EOD and LOD may be different. First, patients with EOD and LOD exhibit different patterns of cognitive impairment: EOD is more associated with language and memory impairment, while LOD is more associated with a decline in executive ability, attention, and processing speed [27,28]. Because many domains of cognitive processing (memory, language, attention) are involved in OI [25], the different patterns of cognitive impairment between EOD and LOD may lead to their difference in OI. Second, accumulating studies have suggested that the patterns of brain abnormalities in EOD and LOD are different, and LOD patients have been found to exhibit more severe hippocampal atrophy than EOD patients. In addition, the hippocampus plays an essential role in OI dysfunction processing, because it is a part of the secondary olfactory cortex and is involved in encoding olfactory information and storing olfactory memory [29,30,31].

Based on the above evidence, we hypothesized that patients with LOD exhibit more severe OI dysfunction than patients with EOD, that they exhibit differences in OI, and that their differences in OI are mediated by their differences in cognitive function. Therefore, the present study aimed to (1) compare OI between patients with EOD and LOD, and (2) explore the relationship between cognitive function and OI in EOD and LOD patients through correlation and mediation analysis. The present study contributes to the rational use of OI for predicting the risk of dementia in LLD and provides a deeper understanding of the different pathological mechanisms between EOD and LOD.

## 2. Materials and Methods

### 2.1. Participants

One hundred and seventy-nine patients with LLD were recruited from the Affiliated Brain Hospital of Guangzhou Medical University (Guangzhou Huiai Hospital), and 189 normal controls (NCs) were recruited from the communities in Guangzhou. All subjects or relevant legal guardians provided written informed consent to participate in the study. The study protocol and assessments were approved by the Ethics Committees of the Affiliated Brain Hospital of Guangzhou Medical University.

Patients with LLD were included in the present study according to the following inclusion criteria: (1) age > 55 years, (2) major depression diagnosed according to DSM-IV criteria, and (3) clinical staging and diagnosis made by at least two neurologists with expertise in dementia, a neuropsychologist, and a geriatric psychiatrist. The exclusion criteria included (1) a history of other major psychiatric disorders, such as bipolar disorder or schizophrenia; (2) physical illnesses that could lead to depressive episodes, such as anaemia or hypothyroidism; (3) neurological disorders, such as brain tumours, Parkinson’s disease, multiple sclerosis, and stroke; (4) current or previous psychiatric symptoms; (5) head injury with loss of consciousness > 30 min; (6) other conditions significantly affecting the sense of smell, including active upper airway/sinus infection or dyspnea at the time of testing, traumatic or congenital olfactory impairment, known nasal tumours or polyps, current or recent (past 6 months) smoking, and alcohol abuse or substance dependence.

For further analyses, patients with LLD were divided into two subgroups: the EOD group (N = 92 cases), whose first depressive episode occurred before age 60, and the LOD group (N = 87 cases), whose depressive episode occurred after age 60 [3].

### 2.2. Clinical Measurements

Demographic information (sex, age, years of education,) and clinical history (age at first onset, duration of illness, duration of depressed mood, number of depressive episodes, lifetime duration of depression, duration of untreated illness) of all subjects was collected at enrolment. The severity of depressive symptoms was assessed using the Geriatric Depression Scale (GDS). All scale assessments were completed by two trained professional psychiatrists who passed a concordance assessment.

### 2.3. Neuropsychological Assessments

Participants underwent comprehensive neuropsychological testing to evaluate vari ous domains of cognition after completing a standard clinical assessment. (1) The Mini-Mental State Examination (MMSE) (score range 0–30) was used to assess global cognitive function. (2) Memory was assessed using the Auditory Verbal Learning Test (AVLT), with AVLT N1–3, AVLT N4, AVLT N5, and AVLT N6 representing immediate recall (score range 0–36), short-term delayed recall (score range 0–12), long-term delayed recall (score range 0–12), and recognition (score range 0–24), respectively. (3) Executive function was assessed using the Stroop Colour and Word Test (Stroop A, Stroop B, Stroop C) and the Trail-Making Test (TMT-A, TMT-B), with scores depending on the number of seconds it took for the subject to complete the test. (4) Language function was assessed using the Boston Naming Test (BNT), with scores ranging from 0–30, and the Verbal Fluency Test (VFT), depending on how many names containing “water” were produced in 60 s. (5) Attention was assessed using the Symbolic Digit Transformation Test (SDMT), with scores depending on the number of symbols filled in correctly within 90 s, and the Digit Span Test (DST), depending on the number of digits recited, ranging from 0–7. (6) Visuospatial ability was assessed using the Clock Drawing Test 4 [CDT4] (ranged from 0–4) and the Rey–Osterrieth Test of Complex Graphics (ROCF) (ranged from 0–36).

### 2.4. Olfactory Assessments

Before the olfactory test, subjects were asked to complete a questionnaire that excluded factors that could affect olfactory function (i.e., history of nasal trauma and related surgery, history of radiation or chemotherapy, nasal congestion, etc.). OI was assessed using the Sniffin’ Sticks Screen 16 test for olfactory assessment [32]. Subjects were asked to smell 16 common odors in order and were asked to select the one image out of four that best matched the odor smelled, of which only one was correct. Subjects were given scores ranging from 0 to 16. The experiment was conducted in a quiet, well-ventilated room free of odors. All participants who completed the neuropsychological assessment were given the OI test on the same day.

### 2.5. Statistical Analyses

The Statistical Package for the Social Sciences Version 26.0 (IBM SPSS 26.0, Chicago, IL, USA) was used to perform statistical analyses. Demographic and clinical variables between the three groups were assessed using chi-square test analysis and one-way ANOVA, and the Mann–Whitney U test was used for comparison if the data did not conform to a normal distribution. The OI and cognitive function of the three groups were compared using analysis of covariance (ANCOVA) adjusted for age, sex, years of education and GDS scores, and multiple comparisons were made using the post hoc least significant difference (LSD) test. The relationship between OI and cognitive function among the three groups was analysed by partial correlation analysis, and control variables included age, sex, years of education, and GDS scores. After utilising the Benjamini–Hochberg procedure, false discovery rate (FDR)-corrected *p* values (i.e., q values) lower than 0.05 were considered statistically significant. Stepwise multiple linear regression was used to analyse the effects of different cognitive domains on olfactory scores, including global cognition (MMSE), memory (AVLT N1–3, AVLT N4, AVLT N5, and AVLT N6), executive function (Stroop C, TMTA, TMTB), language (BNT, VFT), attention (SDMT, DST), and visuospatial skills (CDT4, ROCF). A significance level of *p* < 0.05 was included in the regression model. Mediation analysis was conducted using PROCESS in SPSS to determine whether the association between grouping (EOD/LOD) and OI scores was mediated by changes in neuropsychological indicators. Subsequently, we used the SPSS Process Macro to examine the null hypothesis [33]. The data met the normality and homoscedasticity assumptions. Neuropsychological indicators that were significantly associated with OI were defined as potential variables in the mediation analyses, and a mediation model was created when the following conditions were met: (1) X had a significant effect on Y; (2) X had a significant predictive effect on the mediating variable; (3) the mediator had a significant effect on Y; (4) the effect of X on Y decreased when the mediator entered the model. The purpose of the model is to investigate the total (C) and direct effects (a, b, c′) of the significance between the independent variable (X) and the dependent variable (Y) in each model, as well as the indirect effects (IE) obtained from the product of the coefficients (a*b) [34].

## 3. Results

### 3.1. Demographic and Clinical Characteristics

The demographic and clinical characteristics of the EOD, LOD, and NC groups are shown in Table 1. There were significant differences in sex distribution, age, years of education, and GDS scores among the three groups (*p* < 0.05). Specifically, there were fewer males in the EOD patients than in the NC group, the EOD patients were younger than the NC group, and the LOD patients were older than the NC group. But there was no significant difference in the age of patients with LLD compared to NCs (*p* = 0.58). The EOD and LOD patients had fewer years of education and higher GDS scores than the NC group. In addition, the duration of depression, depressive episodes, and depression lifetime duration of the EOD group were higher than those of the LOD group (see Table 1).

### 3.2. Comparison of Olfactory and Cognitive Functions between Different Groups

There were significant differences in OI scores (*p* = 0.018) (see Figure 1A) and the percentage of OI dysfunction (*p* < 0.001) (see Figure 1B) between the EOD, LOD and NC groups. Specifically, the EOD and NC groups exhibited higher OI scores and a lower proportion of subjects with OI dysfunction than the LOD group (see Figure 1). For the cognitive scores, there were significant differences in MMSE (*p* < 0.001), TMT-A (*p* = 0.03), VFT (*p* = 0.03), and SDMT (*p* = 0.001) scores among the three groups. MMSE scores were higher in the EOD and NC groups than in the LOD, TMT-A scores were greater in the LOD than in the NC group, and VFT scores were higher in the NC than in the EOD group. SDMT scores were smaller in both EOD and LOD patients than in the NC group. (see Table 2).

### 3.3. Correlation Analyses between OI Cognitive Function and Depression

Partial correlation analysis suggested that OI scores were positively correlated with MMSE (r = 0.385, *p* = 0.001, q = 0.008) (see Figure 2A), AVLT N1–3 (r = 0.442, *p* < 0.001, q < 0.001) (see Figure 2B), AVLT N4 (r = 0.396, *p* = 0.002, q = 0.008) (see Figure 2C), AVLT N6 (r = 0.391, *p* = 0.002, q = 0.008) (see Figure 2E), BNT (r = 0.381, *p* = 0.002, q = 0.008) (see Figure 2F), and ROCF (r = 0.347, *p* = 0.006, q = 0.016) (see Figure 2H) in EOD patients. In the NC group, OI scores were correlated with AVLT N1–3 (r = 0.286, *p* < 0.001, q < 0.001), AVLT N4 (r = 0.299, *p* < 0.001, q < 0.001), AVLT N5 (r = 0.203, *p* = 0.008, q = 0.032) (see Figure 2D), and SDMT (r = 0.211, *p* = 0.006, q = 0.032) (see Figure 2G). In the LOD group, there was no significant correlation between OI scores and cognitive scores (*p* > 0.05). In addition, OI scores had no significant correlation with the GDS (*p* > 0.05), disease duration (*p* > 0.05), number of episodes (*p* > 0.05), lifetime duration (*p* > 0.05), or untreated time (*p* > 0.05) in the EOD, LOD and NC groups (see Table 3).

### 3.4. Multiple Linear Regression Analysis

In stepwise multiple regression analysis, OI was most strongly associated with AVLT N1–3 (B = 0.209, 0.001) in the EOD group (R2 = 0.220). Moreover, AVLT N1–3 (B = 0.144, *p* < 0.001), and SDMT (B = 0.049, *p* = 0.005) had a significant effect on the NC group (R2 = 0.1) (see Table 4).

### 3.5. Mediation Analysis

Considering the close relationship between cognitive function and OI scores, mediation analysis was used to further explore their relationship. Overall, we found two mediating models (Figure 3). First, the total effect of the EOD/LOD group on OI was β = −1.73 (*p* < 0.01, lower limit of confidence interval (LLCI) = −2.54, upper limit of confidence interval (ULCI) = −0.92). The indirect effect of each group on OI via AVLT N1–3 was β = −0.31 (LLCI = −0.69, ULCI = −0.04) (Figure 3A). Next, the total effect of the EOD/LOD group on OI scores was β = −1.72 (*p* < 0.01, LLCI = −2.54, ULCI = −0.91). The indirect effect of each group on OI scores through BNT was −0.31 (LLCI = −0.66, ULCI = −0.06) (Figure 3B). The above results suggest that the difference in OI scores between the EOD and LOD groups was partially mediated by AVLT N1–3 and BNT. Other cognitive function factors did not exhibit a significant mediating effect on the difference in OI between the EOD and LOD groups (*p* > 0.05).

## 4. Discussion

The present study first explored the difference in OI between EOD and LOD patients and explored the relationship between OI dysfunction and cognitive impairment by using mediation analysis. The main findings were as follows. (1) LOD patients exhibited lower OI scores than EOD patients and normal controls (NCs). Additionally, the LOD patients exhibited a higher percentage of OI dysfunction than the EOD patients. (2) OI scores were associated with global cognition, memory, language, and visuospatial ability in the EOD group but not with any cognitive score in the LOD group. (3) AVLT N1–3 and BNT exhibited a partially mediating effect on the difference in OI scores between the EOD and LOD groups.

Consistent with our previous studies, the present results suggested that patients with LLD exhibit OI dysfunction compared with normal controls [24,25]. Croy et al. proposed that olfactory dysfunction in depression may lead to a decrease in the attention and turnover rate of olfactory receptor neurons, and a decrease in olfactory bulb volume may reduce the signal from the olfactory bulb to the olfactory cortex, and exacerbate depressive symptoms [35]. Olfactory threshold dysfunction has been consistently reported in patients with depression [20,36], but whether they exhibit OI dysfunction is still in dispute. Khil et al. pointed out that OI dysfunction is related to depression in patients with first-time high symptom severity and a severe disease course, and found that the prevalence of OI dysfunction in depression patients and normal controls was similar (approximately 15%) [37]. The present study suggested that OI dysfunction is also associated with cognitive impairment in LLD patients, which is consistent with our previous studies [24,25]. In addition, OI dysfunction is also associated with more severe structural and functional brain abnormalities in LLD, which suggests that OI dysfunction may indicate an elevated risk of dementia in LLD. However, it remains unclear whether patients with EOD and LOD may exhibit different OI, which introduces a potential confounding factor for using OI to predict dementia risk in LLD.

Our study found that LOD patients exhibited worse OI than EOD patients and that OI was associated with cognitive impairment in EOD patients but not in LOD patients. On the one hand, the pathogenesis of EOD is mainly associated with the disturbance of cortisol rhythm caused by chronic stress, and the negative feedback of the HPA axis causes elevated cortisol levels, leading to hippocampal atrophy and cognitive impairment [38,39]. On the other hand, LOD is more inclined to the age-related cerebrovascular lesion hypothesis, where microvascular dysfunction leads to reduced local cerebral perfusion and ischaemia in the corresponding areas (cortex, grey matter, hippocampus, etc.), causing corresponding emotional and cognitive impairment [40,41]. The depression-executive function hypothesis has also been widely reported in LOD, which may be related to frontal striatum dysfunction and subcortical white matter damage [42,43]. Papazacharias et al. assumed that the potential mechanisms of EOD and LOD in the relationship between depression and cognitive impairment may be different. Although the association between EOD and cognitive decline may be best understood as involving glucocorticoid function, neurological changes are thought to contribute to cognitive deficits and depressive symptoms in LOD [44]. Moreover, LOD is more likely to be a prodromal symptom of AD [45]. Considering that OI dysfunction is associated with dementia risk and hippocampal dysfunction, the worse OI in LOD compared with EOD is consistent with the previous hypothesis.

The present study suggested that the OI difference between EOD and LOD is mediated by language and attention. OI processes are not only associated with odor perception, but also involve many higher cognitive processes, such as semantic memory, working memory, verbal fluency, processing speed and reasoning ability [46,47,48]. Additionally, many brain regions of OI (entorhinal cortex, hippocampus, insula, orbitofrontal cortex, piriform cortex, etc.) [49,50,51] also have some overlap with the language and attention network. Thus, whether the OI difference between EOD and LOD is regulated by the dysfunction of these overlapped brain regions needs to be further verified.

There are several limitations in the present study. First, the current study was cross-sectional and did not dynamically observe cognitive and olfactory changes. Follow-up studies are needed to explore whether OI dysfunction exhibits different predictive powers for dementia in patients with EOD and LOD. Second, the diagnoses of EOD and LOD were based on self-reports and the retrospective provision of medical history, which may have produced recall bias. Third, although GDS scores and other covariates were considered in the analyses, most LLD patients were under variable doses of antidepressants, and the confounding effect of medication was not excluded. Fourth, odor identification (OI) dysfunction is considered as a noninvasive, inexpensive tool for dementia prediction, but we must consider that its specificity is relatively poor, because patients with Parkinson’s disease, Lewy body dementia may also have impaired olfaction and depression, and we need to combine PET-CT and cerebrospinal fluid markers to increase the specificity in future studies. Fifth, although age was used as a covariate, the significant age difference between the three groups is one of the limitations of this study. Sixth, olfactory thresholds and olfactory discrimination were not included in the present assessments. Future studies will conduct a comprehensive assessment of olfactory function that includes olfactory identification, olfactory thresholds, and olfactory discrimination to better clarified the difference between EOD and LOD. Seventh, it remains unclear which brain abnormalities were involved in the different OI between EOD and LOD, and future studies that include neuroimaging data are needed to clarify this question. Eighth, the present study did not strictly distinguish between acute and remitted stages of LLD, but the GDS was not associated with OI scores, and the GDS was included as a covariate in all analyses to minimize the confounding effect of depressive stage on OI.

## 5. Conclusions

The present study suggests that LOD patients exhibit worse OI than EOD patients, and their differences in memory and language function contribute to their differences in OI. The underlying mechanism of the different OI between EOD and LOD patients needs to be further explored, and the age of the first episode of depression needs to be considered when using OI dysfunction as a predictor of dementia risk in LLD patients.

## Figures and Tables

**Figure 1 brainsci-12-00276-f001:**
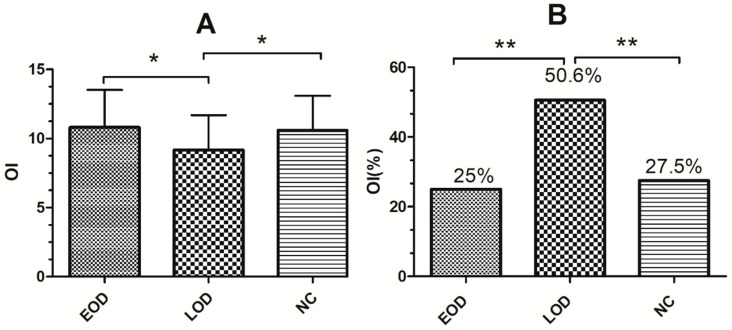
Comparison of the OI scores (**A**) and the percentage of OI dysfunction (**B**) among the EOD, LOD and NC groups. OI, odor identification; OI (%), percentage of OI dysfunction. EOD means patients with early onset depression; LOD means patients with late onset depression; NC means normal controls. * *p* < 0.05; ** *p* < 0.01.

**Figure 2 brainsci-12-00276-f002:**
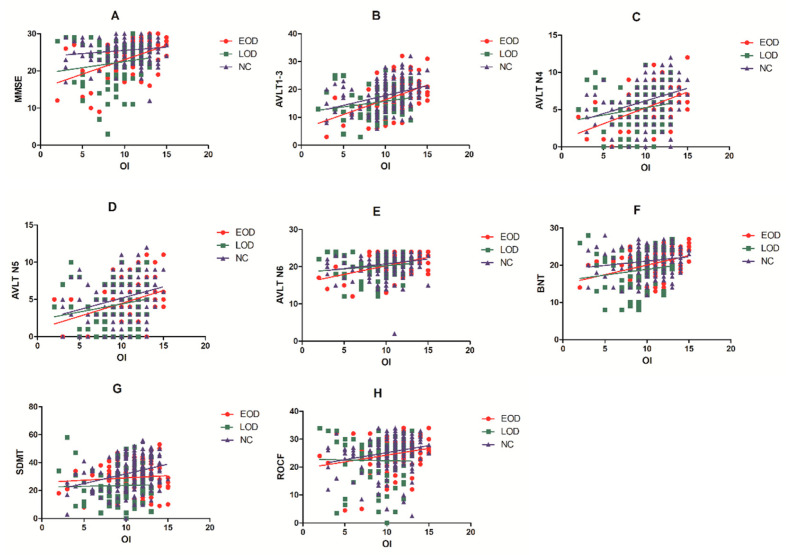
Correlational analyses between cognitive function and OI in EOD, LOD and NC groups. (**A**) OI was associated with MMSE in EOD (r = 0.385, *p* = 0.001, q = 0.008) but not in NC and LOD. (**B**) OI was associated with AVLT N1–3 in EOD (r = 0.442, *p* < 0.001, q < 0.001) and NC (r = 0.286, *p* < 0.001, q < 0.001) but not in LOD. (**C**) OI was associated with AVLT N4 in EOD (r = 0.396, *p* = 0.002, q = 0.008) and NC (r = 0.299, *p* < 0.001, q < 0.001) but not in LOD. (**D**) OI was associated with AVLT N5 in NC (r = 0.203, *p* = 0.008, q = 0.032) but not in EOD and LOD. (**E**) OI was associated with AVLT N6 in EOD (r = 0.391, *p* = 0.002, q = 0.008) but not in LOD and NC. (**F**) OI was associated with BNT in EOD (r = 0.381, *p* = 0.002, q = 0.008) but not in LOD and NC. (**G**) OI was associated with SDMT in NC (r = 0.211, *p* = 0.006, q = 0.032) but not in EOD and LOD. (**H**) OI was associated with ROCF in EOD (r = 0.347, *p* = 0.006, q = 0.016) but not in LOD and NC. OI, odor identification; MMSE, Mini-Mental State Examination.

**Figure 3 brainsci-12-00276-f003:**
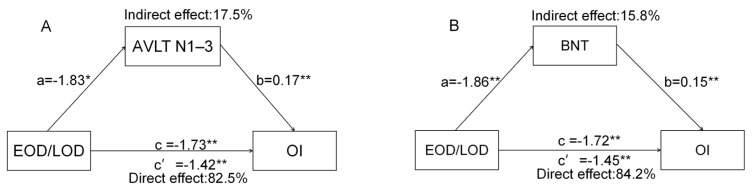
The mediating effect of cognitive function on the OI scores of EOD/LOD groups. (**A**) AVLT N1–3 mediated the difference in OI scores of EOD/LOD groups. (**B**) BNT mediated the difference in OI scores of EOD/LOD groups. * *p* < 0.05; ** *p* < 0.01; EOD means patients with early-onset depression; LOD means patients with late-onset depression. AVLT N1–3, Auditory Verbal Learning Test Immediate recall; BNT, Boston Naming Test; OI, odor identification.

**Table 1 brainsci-12-00276-t001:** Demographic and clinical data of EOD, LOD and NC groups.

	EOD (*n* = 92)	LOD (*n* = 87)	NC (*n* = 189)	*F*/t/χ^2^	*p*	Post hoc
Male (%)	16 (17.4%)	24 (27.6%)	70 (37.0%)	11.69	0.003	A < C
Age	64.85 ± 5.02	70.92 ± 6.91	67.29 ± 7.49	20.65	0.001	B > C > A
Years of education (years)	9.04 ± 3.66	8.38 ± 4.11	10.38 ± 3.36	10.30	0.001	C > A, B
Number of people taking antidepressants	81	68	NA	3.13	0.07	——
Age at first onset	53.50 (49.00–57.00)	66.00 (62.00–73.00)	NA	−10.00	0.001	A < B
Disease duration (years)	12.10 ± 12.71	4.38 ± 5.47	NA	22.90	0.001	A > B
Number of episodes	3.06 ± 3.75	1.54 ± 1.26	NA	15.98	0.002	A > B
Lifetime duration (years)	9.29 ± 8.90	2.75 ± 2.82	NA	42.91	0.001	A > B
Untreated time (years)	3.22 ± 7.77	1.56 ± 3.28	NA	5.70	0.11	——
GDS	5.55 ± 4.14	4.35 ± 3.73	2.28 ± 2.30	28.37	0.001	A > B > C

GDS, Geriatric Depression Scale; EOD, early onset depression; LOD, late onset depression; NC, normal controls. For post-hoc comparison, A means EOD; B group means LOD group; C means NC group. NA, not applicable.

**Table 2 brainsci-12-00276-t002:** Comparison of olfactory and cognitive functions between EOD, LOD and NC.

	EOD (*n* = 92)	LOD (*n* = 87)	NC (*n* = 189)	*F*	*p*	Post hoc
OI	10.87 ± 2.78	9.29 ± 2.62	10.75 ± 2.46	4.10	0.018	A, C < B
OI dysfunction (%)	23(25.0%)	44(50.6%)	52(27.5%)	17.50	0.001	A, C < B
Global cognition						
MMSE	23.72 ± 4.69	22.10 ± 5.56	25.78 ± 2.81	9.94	<0.001	A, C > B
Memory						
AVLT N1–3	17.69 ± 6.13	15.82 ± 4.80	18.51 ± 4.73	0.794	0.45	——
AVLT N4	5.83 ± 2.46	5.08 ± 2.59	6.52 ± 2.33	1.44	0.24	——
AVLT N5	4.91 ± 2.93	4.17 ± 2.75	5.51 ± 2.66	0.34	0.72	——
AVLT N6	20.68 ± 2.62	20.38 ± 2.43	21.07 ± 2.66	0.40	0.67	——
Executive function						
TMTA	61.53 ± 21.03	68.25 ± 28.75	51.80 ± 21.54	3.52	0.03	B > C
TMTB	76.77 ± 34.13	91.69 ± 36.76	71.16 ± 31.13	2.84	0.06	——
Stroop A	33.36 ± 9.20	35.37 ± 9.43	31.27 ± 8.76	2.17	0.12	——
Stroop B	47.44 ± 15.77	45.97 ± 14.27	43.76 ± 13.77	1.08	0.34	——
Stroop C	92.73 ± 30.99	94.04 ± 33.71	88.96 ± 36.59	0.71	0.49	——
Language						
BNT	20.68 ± 3.82	19.25 ± 4.89	21.44 ± 3.37	2.85	0.06	——
VFT	7.92 ± 3.18	8.02 ± 3.73	9.55 ± 3.67	3.64	0.03	C > A
Attention						
SDMT	28.55 ± 10.33	23.61 ± 12.72	33.15 ± 10.68	7.49	0.001	A, B < C
DST	5.03 ± 1.04	4.80 ± 1.27	5.18 ± 1.07	0.35	0.71	——
Visuospatial skill						
ROCF	24.98 ± 5.81	22.81 ± 7.90	25.83 ± 5.31	0.77	0.46	——
CDT4	3.52 ± 0.73	3.25 ± 0.82	3.53 ± 0.77	2.56	0.08	——

MMSE, Mini-Mental State Examination; AVLT N1–3, Auditory Verbal Learning Test Immediate recall; AVLT N4, Auditory Verbal Learning Test Short-term delayed recall; AVLT N5, Auditory Verbal Learning Test Long-term delayed recall; AVLT N6, Auditory Verbal Learning Test Recognition; TMT, Trail-Making Test; Stroop, The Stroop Colour and Word Test; BNT, Boston Naming Test; VFT, Verbal Fluency Test; SDMT, Symbol-Digit; DST, digit span test; ROCF, Rey–Osterrieth Complex; CDT, Clock Drawing Task; A adjusted for age, sex, years of education and GDS scores. For post hoc comparison, A indicates the EOD group; B indicates the LOD group; C indicates the NC group.

**Table 3 brainsci-12-00276-t003:** Correlation analyses between OI and cognitive function in EOD, LOD and NC groups.

−	EOD			LOD			NC		
	r	*p*	q	r	*p*	q	r	*p*	q
MMSE	0.385	0.001	0.008	0.095	0.453	0.906	0.030	0.692	0.738
AVLT N1–3	0.442	*p* < 0.001	q < 0.001	0.123	0.368	0.981	0.286	*p* < 0.001	q < 0.001
AVLT N4	0.396	0.002	0.008	0.070	0.606	0.693	0.299	*p* < 0.001	q < 0.001
AVLT N5	0.238	0.065	0.130	0.091	0.507	0.737	0.203	0.008	0.032
AVLT N6	0.391	0.002	0.008	0.019	0.892	0.892	0.175	0.022	0.059
TMTA	0.185	0.163	0.261	0.053	0.636	0.678	0.048	0.214	0.285
TMTB	−0.008	0.950	0.950	−0.142	0.306	1.000	−0.062	0.423	0.483
Stroop A	−0.269	0.034	0.078	−0.251	0.065	1.000	−0.181	0.018	0.058
Stroop B	−0.198	0.123	0.219	−0.091	0.515	0.687	−0.100	0.197	0.287
Stroop C	−0.022	0.868	0.992	−0.078	0.578	0.711	−0.021	0.788	0.788
BNT	0.381	0.002	0.008	0.145	0.286	1.000	0.115	0.135	0.240
VFT	−0.015	0.906	0.966	−0.093	0.496	0.794	0.130	0.091	0.182
SDMT	0.029	0.822	1.000	−0.131	0.353	1.000	0.211	0.006	0.032
DST	0.162	0.209	0.304	0.205	0.131	1.000	0.087	0.274	0.337
ROCF	0.347	0.006	0.016	−0.097	0.477	0.848	0.149	0.052	0.119
CDT4	0.109	0.399	0.532	0.119	0.381	0.871	0.106	0.167	0.267

MMSE, Mini-Mental State Examination; AVLT N1–3, Auditory Verbal Learning Test Immediate recall; AVLT N4, Auditory Verbal Learning Test Short-term delayed recall; AVLT N5, Auditory Verbal Learning Test Long-term delayed recall; AVLT N6, Auditory Verbal Learning Test Recognition; TMT, Trail-Making Test; Stroop, The Stroop Colour and Word Test; BNT, Boston Naming Test; VFT, Verbal Fluency Test; SDMT, Symbol-Digit; DST, digit span test; ROCF, Rey–Osterrieth Complex; CDT, Clock Drawing Task. The false discovery rate was adjusted using Benjamini–Hochberg procedure, and q-values are reported; q-values less than 0.05 were considered significant. Adjusted for age, sex, years of education and GDS scores. EOD means patients with early onset depression; LOD means patients with late onset depression; NC means normal controls.

**Table 4 brainsci-12-00276-t004:** Multiple linear regression analyses of the association between OI and cognitive function in EOD and NC groups.

	R^2^	Variables	B	β	t	*p*	95% CI
EOD	0.220	AVLT N1–3	0.209	0.469	4.805	0.001	(0.123, 0.296)
NC	0.168	AVLT N1–3	0.144	0.276	3.707	0.001	(0.068, 0.221)
		SDMT	0.049	0.211	2.838	0.005	(0.015, 0.083)

MMSE, Mini-Mental State Examination; AVLT N1–3, Auditory Verbal Learning Test Immediate recall; SDMT, Symbol-Digit; EOD means patients with early onset depression; LOD means patients with late onset depression; NC means normal controls.

## Data Availability

Data are available upon request to the authors.

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
