# Peer review of "Differences in Odor Identification in Early-Onset and Late-Onset Depression"

_brainsci, 2022, doi:10.3390/brainsci12020276_

Round 1

Reviewer 1 Report

My impressions and general overview: 

  1. In the present study, odor identification was compared between early-onset and late-onset depression.
  2. The paper uses clear, unambiguous, and professional language throughout.
  3. In this paper, there is sufficient background information to show how the work fits into the broader field of knowledge. 
  4. In general, the paper is self-contained and includes all results relevant to the hypothesis. 

Experimental design:

The paper includes a total of 179 patients (EOD subjects = 92 and LOD subjects = 87) with late-life depression (LLD) and 189 normal controls were recruited. Participants underwent clinical assessment, olfactory testing, and comprehensive neuropsychological assessment. The Odor Identification (OI) scores were compared between early-onset depression (EOD) patients and late-onset depression (LOD) patients, and correlation analyses and mediation analyses were used to explore the relationship between OI and cognition.

The paper reports the following major findings:

  1. Positive findings are:
    1. Both EOD patients and LOD patients exhibited lower OI scores compared with normal controls (NC).
    2. In comparison with the EOD group, the LOD group had lower OI scores and a higher percentage of OI dysfunction.
    3. In the EOD group, OI scores were associated with global cognition, memory, executive function, language, and visuospatial ability (P<0.05).
    4. The difference in OI scores between the EOD and LOD groups was partially mediated by the scores of the Auditory Verbal Learning Test Immediate recall and the Boston Naming Test.
  2. Negative findings:
    1. In the LOD group, OI scores were not associated with any cognitive score (P>0.05).

Strengths of the paper

  1. The paper includes clearly describes the introduction, methods, and major findings.
  1. The paper performs sound statistical analysis and reports the details enough to replicate the study.
  2. Conclusions are supported by the evidence presented in the paper.
  3. A knowledge gap exists regarding odor identification in early-onset and late-onset depression, and the paper presents novel findings relevant to the field.

Weaknesses

  1. Typographical and grammatical errors: For example, in the structured abstract with the numerical indexing. However, index number 3 is missing from the abstract. Proofreading can easily correct minor grammatical errors.
  2. As the paper compares multiple tests and results. However, these multiple statistical tests were not corrected for the multiple comparisons. I would suggest correcting the p-values for the multiple comparisons to prevent false-positive results. Although it won't affect negative findings, it will rule out false-positive findings. Benjamini and Hochberg's false discovery rate (FDR) correction method will be useful here for correcting the FDR. In order to establish the reliability of the results, it is vital to correct the p-values.

Overall comment: The paper is generally well written and has a logical flow. It provides enough details for reproducibility and many of its methods are sound.

Reviewer 2 Report

The main aim of the current study was to compare odor identification function between patients in which the first depressive episode occurs before (EOD) or after (LOD) age 60. Furthermore, the effect of various cognitive functions on this association have been tested. The number of participants and cognitive tests are impressive, however some particular issues are need to be addressed:

  1. The significant difference of age between the three groups is a major limitation of this study, despite of the adjustment for age. The authors must address it in their manuscript. [ref 1-2]
  2. Lines 347-348: “most LLD were under variable doses of antidepressants, and the confounding effect of medication was not excluded”. This is an additional major limitation, it should be report how many subjects are taking antidepressants in each group.
  3. Some details regarding the mediation analysis are missing: i. Is the data meet with the normality and homoscedasticity assumptions?   ii. Did the authors use Sobel test or bootstrapping? [ref 3]

Minors:

  1. MoCA detected more cognitive impairments than MMSE in depression, thus the choice in MMSE should be address or add as part of the study limitations.
  2. Line 76: I assume it is a typo: EOD, not LOD:

“Based on the above evidence, we hypothesized that patients with LOD exhibit more severe OI dysfunction compared with patients with LOD exhibit differences OI, and their difference of OI is mediated by their difference of cognitive function.”

Finally, whilst the content is clear, the authors should get editing help from someone with full professional proficiency in English. The grammatical errors distract from the overall quality of the work.

Reference:

  1. Zhang C, Wang X. Initiation of the age-related decline of odor identification in humans: A meta-analysis. Ageing Res Rev. 2017 Nov;40:45-50. doi: 10.1016/j.arr.2017.08.004. Epub 2017 Aug 19. PMID: 28830800.

2.Xu L, Liu J, Wroblewski KE, McClintock MK, Pinto JM. Odor Sensitivity Versus Odor Identification in Older US Adults: Associations With Cognition, Age, Gender, and Race. Chem Senses. 2020 May 21;45(4):321-330. doi: 10.1093/chemse/bjaa018. PMID: 32406505; PMCID: PMC7320224.

  1. Abu-Bader, Soleman and Jones, Tiffanie Victoria, Statistical Mediation Analysis Using the Sobel Test and Hayes SPSS Process Macro (March 6, 2021). International Journal of Quantitative and Qualitative Research Methods.

Reviewer 3 Report

The paper is interesting and presents a useful characterization of the olfactory function in subjects with late life depression, a condition whose prevalence is rapidly increasing and possibly driving to dementia-like forms within a relatively short time frame.

Therefore, the paper has important scientific and clinical consequences, confirming the importance of the olfactory assessment as a useful screening means for a various range of clinical conditions.

The sample size is large enough to draw robust conclusions, therefore the results obtained are trustworthy overall.

I just have a couple of minor remarks before judging the paper as acceptable:

  1. The methodology, albeit reliable, could have been more exhaustive. I am aware of the fact that, in order to investigate such high number of subjects, one might need a very agile methodology, and the full Sniffin' Sticks Extended Test battery could have been time consuming; however, the use of Threshold and Discrimination Tests could have allowed the scientists to better understand the reasons why such changes occur in terms of olfactory perception (e.g., see, for other conditions, Muratori et al., Journal of Autism and Developmental Disorders, 2017; or Tonacci et al., International Journal of Cardiology, 2014, or Journal of Biomedical Science, 2013).
  2. The olfactory assessment is a pretty promising approach to study various disorders related to neurology/psychiatry or neuroscience in general. However, its results are often affected by a very high sensitivity and low specificity. The authors should discuss on this, pointing out the pros and cons of this approach.
  3. Another key issue of the olfactory assessment, especially when it comes to the identification test, is represented by cultural adaptation. How did you check the cultural compliance with a population pretty different than the European/Western one on which the Sniffin' Sticks have been more extensively applied?
  4. Finally, a thorough revision of the English language and grammar should be performed, possibly supported by a native proofreader, which should also check the compliance with the Journal's format.
